# Coefficients of a Comprehensive Subclass of Meromorphic Bi-Univalent Functions Associated with the Faber Polynomial Expansion

**Hari Mohan Srivastava** [1,2,3,4,*] , **Ahmad Motamednezhad** [5] **and Safa Salehian** [6]

1 Department of Mathematics and Statistics, University of Victoria, Victoria, BC V8W 3R4, Canada
2 Department of Medical Research, China Medical University Hospital, China Medical University, Taichung 40402, Taiwan
3 Department of Mathematics and Informatics, Azerbaijan University, 71 Jeyhun Hajibeyli Street, Baku AZ1007, Azerbaijan
4 Section of Mathematics, International Telematic University Uninettuno, I-00186 Rome, Italy
5 Faculty of Mathematical Sciences, Shahrood University of Technology, Shahrood P.O. Box 316-36155, Iran; a.motamedne@gmail.com
6 Department of Mathematics, Gorgan Branch, Islamic Azad University, Gorgan P.O. Box 717, Iran; s.salehian84@gmail.com
* Correspondence: harimsri@math.uvic.ca

**Abstract:** In this paper, we introduce a new comprehensive subclass $\Sigma_B(\lambda, \mu, \beta)$ of meromorphic bi-univalent functions in the open unit disk $\mathbb{U}$. We also find the upper bounds for the initial Taylor-Maclaurin coefficients $|b_0|$, $|b_1|$ and $|b_2|$ for functions in this comprehensive subclass. Moreover, we obtain estimates for the general coefficients $|b_n|$ ($n \geqq 1$) for functions in the subclass $\Sigma_B(\lambda, \mu, \beta)$ by making use of the Faber polynomial expansion method. The results presented in this paper would generalize and improve several recent works on the subject.

**Keywords:** analytic functions; univalent and bi-univalent functions; meromorphic bi-univalent functions; coefficient estimates; Faber polynomial expansion; meromorphic bi-Bazilevič functions of order $\beta$ and type $\mu$; meromorphic bi-starlike functions of order $\beta$

## 1. Introduction

Let $\mathcal{A}$ denote the class of functions $f$ of the form:

$$f(z) = z + \sum_{n=2}^{\infty} a_n z^n, \tag{1}$$

which are analytic in the open unit disk

$$\mathbb{U} = \{z : z \in \mathbb{C} \quad \text{and} \quad |z| < 1\}.$$

We also let $\mathcal{S}$ be the class of functions $f \in \mathcal{A}$ which are univalent in $\mathbb{U}$.

It is well known that every function $f \in \mathcal{S}$ has an inverse $f^{-1}$, which is defined by

$$f^{-1}(f(z)) = z \qquad (z \in \mathbb{U})$$

and

$$f(f^{-1}(w)) = w \qquad \left(|w| < r_0(f); \ r_0(f) \geqq \frac{1}{4}\right).$$

If $f$ and $f^{-1}$ are univalent in $\mathbb{U}$, then $f$ is said to be bi-univalent in $\mathbb{U}$. We denote by $\sigma_\mathcal{B}$ the class of bi-univalent functions in $\mathbb{U}$. For a brief history and interesting examples of functions in the class $\sigma_\mathcal{B}$, see the pioneering work [1]. In fact, this widely-cited work

by Srivastava et al. [1] actually revived the study of analytic and bi-univalent functions in recent years, and it has also led to a flood of papers on the subject by (for example) Srivastava et al. [2–14] and by others [15,16].

In this paper, let $\Sigma$ be the family of meromorphic univalent functions $f$ of the following form:

$$f(z) = z + b_0 + \sum_{n=1}^{\infty} \frac{b_n}{z^n}, \tag{2}$$

which are defined on the domain

$$\Delta = \{z : z \in \mathbb{C} \quad \text{and} \quad 1 < |z| < \infty\}.$$

Since a function $f \in \Sigma$ is univalent, it has an inverse $f^{-1}$ that satisfies the following relationship:

$$f^{-1}(f(z)) = z \qquad (z \in \Delta)$$

and

$$f(f^{-1}(w)) = w \qquad (M < |w| < \infty; \ M > 0).$$

Furthermore, the inverse function $f^{-1}$ has a series expansion of the form [17]:

$$g(w) = f^{-1}(w) = w + \sum_{n=0}^{\infty} \frac{B_n}{w^n} \qquad (M < |w| < \infty).$$

A function $f \in \Sigma$ is said to be meromorphic bi-univalent if both $f$ and $f^{-1}$ are meromorphic univalent in $\Delta$. The family of all meromorphic bi-univalent functions in $\Delta$ of the form (2) is denoted by $\Sigma_{\mathcal{M}}$. A simple calculation shows that (see also [18,19])

$$g(w) = f^{-1}(w) = w - b_0 - \frac{b_1}{w} - \frac{b_2 + b_0 b_1}{w^2} - \cdots . \tag{3}$$

Moreover, the coefficients of $g = f^{-1}$ can be given in terms of the *Faber polynomial* [20] (see also [21–23]) as follows:

$$g(w) = f^{-1}(w) = w - b_0 - \sum_{n=1}^{\infty} \frac{1}{n} K_{n+1}^n \frac{1}{w^n} \qquad (w \in \Delta), \tag{4}$$

where

$$K_{n+1}^n = n b_0^{n-1} b_1 + n(n-1) b_0^{n-2} b_2 + \frac{1}{2} n(n-1)(n-2) b_0^{n-23}(b_3 + b_1^2)$$
$$+ \frac{n(n-1)(n-2)(n-3)}{3!} b_0^{n-4}(b_4 + 3b_1 b_2) + \sum_{j \geq 5} b_0^{n-j} V_j$$

and $V_j$ (with $5 \leqq j \leqq n$) is a homogeneous polynomial of degree $j$ in the variables $b_1, b_2, \cdots, b_n$.

Estimates on the coefficients of meromorphic univalent functions were widely investigated in the literature. For example, Schiffer [24] obtained the estimate $|b_2| \leq 2/3$ for meromorphic univalent functions $f \in \Sigma$ with $b_0 = 0$ and Duren [25] proved that

$$|b_n| \leqq \frac{2}{n+1} \qquad \left(f \in \Sigma; \ b_k = 0; \ 1 \leqq k < \frac{n}{2}\right).$$

Many researchers introduced and studied subclasses of meromorphic bi-univalent functions (see, for instance, Janani et al. [26], Orhan et al. [27] and others [28–30]).

Recently, Srivastava et al. [31] introduced a new class $\Sigma_{B^*}(\lambda, \beta)$ of meromorphic bi-univalent functions and obtained the estimates on the initial Taylor–Maclaurin coefficients $|b_0|$ and $|b_1|$ for functions in this class.

**Definition 1** (see [31]). *A function $f \in \Sigma_{\mathcal{M}}$, given by* (2), *is said to be in the class $\Sigma_{B^*}(\lambda, \beta)$ ($\lambda \geqq 1$; $0 \leqq \beta < 1$), if the following conditions are satisfied:*

$$\Re\left(\frac{z(f'(z))^{\lambda}}{f(z)}\right) > \beta$$

*and*

$$\Re\left(\frac{w(g'(w))^{\lambda}}{g(w)}\right) > \beta,$$

*where the function $g$, given by* (3) *is the inverse of $f$ and $z, w \in \Delta$.*

**Theorem 1** (see [31]). *Let the function $f \in \Sigma_{\mathcal{M}}$, given by* (2), *be in the class $\Sigma_{B^*}(\lambda, \beta)$. Then,*

$$|b_0| \leqq 2(1-\beta) \qquad and \qquad |b_1| \leqq \frac{2(1-\beta)\sqrt{4\beta^2 - 8\beta + 5}}{1+\lambda}.$$

In this paper, we introduce a new comprehensive subclass $\Sigma_B(\lambda, \mu, \beta)$ of the meromorphic bi-univalent function class $\Sigma_{\mathcal{M}}$. We also obtain estimates for the initial Taylor–Maclaurin coefficients $b_0$, $b_1$ and $b_2$ for functions in this subclass. Furthermore, we find estimates for the general coefficients $b_n (n \geq 1)$ for functions in this comprehensive subclass $\Sigma_B(\lambda, \mu, \beta)$ by using the Faber polynomials [20]. Our results for the meromorphic bi-univalent function subclass $\Sigma_B(\lambda, \mu, \beta)$ would generalize and improve some recent works by Srivastava et al. [31], Hamidi et al. [32] and Jahangiri et al. [33] (see also the recent works [34,35]).

## 2. Preliminary Results

For finding the coefficients of functions belonging to the function class $\Sigma_B(\lambda, \mu, \beta)$, we need the following lemmas and remarks.

**Lemma 1** (see [21,22]). *Let $f$ be the function given by*

$$f(z) = z + b_0 + \frac{b_1}{z} + \frac{b_2}{z^2} + \cdots$$

*be a meromorphic univalent function defined on the domain $\Delta$. Then, for any $\rho \in \mathbb{R}$, there are polynomials $K_n^{\rho}$ such that*

$$\left(\frac{f(z)}{z}\right)^{\rho} = 1 + \sum_{n=1}^{\infty} \frac{K_n^{\rho}(b_0, b_1, \cdots, b_{n-1})}{z^n},$$

*where*

$$K_n^{\rho}(b_0, b_1, \cdots, b_{n-1}) = \rho b_{n-1} + \frac{\rho(\rho-1)}{2}D_n^2 + \frac{\rho!}{(\rho-3)!3!}D_n^3 + \cdots + \frac{\rho!}{(\rho-n)!n!}D_n^n$$

*and*

$$D_n^k(x_1, x_2, \cdots, x_{n-k+1}) = \sum \frac{k!(x_1)^{\mu_1} \cdots (x_{n-k+1})^{\mu_{n-k+1}}}{\mu_1! \cdots \mu_{n-k+1}!},$$

*in which the sum is taken over all non-negative integers $\mu_1, \cdots, \mu_{n-k+1}$ such that*

$$
\begin{cases}
\mu_1 + \mu_2 + \cdots + \mu_{n-k+1} = k \\[2mm]
\mu_1 + 2\mu_2 + \cdots + (n-k+1)\mu_{n-k+1} = n.
\end{cases}
$$

The first three terms of $K_n^\rho$ are given by

$$
K_1^\rho(b_0) = \rho b_0,
$$

$$
K_2^\rho(b_0, b_1) = \rho b_1 + \frac{\rho(\rho-1)}{2} b_0^2
$$

and

$$
K_3^\rho(b_0, b_1, b_2) = \rho b_2 + \rho(\rho-1)b_0 b_1 + \frac{\rho(\rho-1)(\rho-2)}{3!} b_0^3.
$$

**Remark 1.** *In the special case when*

$$
b_0 = b_1 = \cdots = b_{n-1} = 0,
$$

*it is easily seen that*

$$
K_i^\rho(b_0, \cdots, b_{i-1}) = 0 \qquad (1 \leqq i \leqq n)
$$

*and*

$$
K_{n+1}^\rho(b_0, b_1, \cdots, b_n) = \rho b_n.
$$

**Lemma 2** (see [21,22])**.** *Let $f$ be the function given by*

$$
f(z) = z + b_0 + \frac{b_1}{z} + \frac{b_2}{z^2} + \cdots
$$

*be a meromorphic univalent function defined on the domain $\Delta$. Then, the Faber polynomials $F_n$ of $f(z)$ are given by*

$$
\frac{z f'(z)}{f(z)} = 1 + \sum_{n=1}^{\infty} \frac{F_n(b_0, b_1, \cdots, b_{n-1})}{z^n}, \tag{5}
$$

*where $F_n(b_0, b_1, \cdots, b_{n-1})$ is a homogeneous polynomial of degree $n$.*

**Remark 2** (see [36])**.** *For any integer $n \geqq 1$, the polynomials $F_n(b_0, b_1, \cdots, b_{n-1})$ are given by*

$$
F_n(b_0, b_1, \cdots, b_{n-1}) = \sum_{i_1 + 2i_2 + \cdots + ni_n = n} A_{(i_1, i_2, \cdots, i_n)} b_0^{i_1} b_1^{i_2} \cdots b_{n-1}^{i_n},
$$

*where*

$$
A_{(i_1, i_2, \cdots, i_n)} := (-1)^{n+2i_1+3i_2+\cdots+(n+1)i_n} \frac{(i_1 + i_2 + \cdots + i_n - 1)! \, n}{i_1! \, i_2! \, \cdots \, i_n!}.
$$

The first three terms of $F_n$ are given by

$$
F_1(b_0) = -b_0,
$$

$$
F_2(b_0, b_1) = b_0^2 - 2b_1
$$

and

$$
F_3(b_0, b_1, b_2) = -b_0^3 + 3b_0 b_1 - 3b_2.
$$

**Remark 3.** *In the special case when $b_0 = b_1 = \cdots = b_{n-1} = 0$, it is readily observed that*

$$
F_i(b_0, \cdots, b_{i-1}) = 0 \qquad (1 \leqq i \leqq n)
$$

*and*

$$F_{n+1}(b_0, b_1, \cdots, b_n) = (-1)^{2n+3}(n+1)b_n = -(n+1)b_n.$$

**Lemma 3.** *Let $f$ be the function given by*

$$f(z) = z + b_0 + \frac{b_1}{z} + \frac{b_2}{z^2} + \cdots$$

*be a meromorphic univalent function defined on the domain $\Delta$. Then, for $\lambda \geqq 1$ and $\mu \geqq 0$,*

$$\left(\frac{zf'(z)}{f(z)}\right)^\lambda \left(\frac{f(z)}{z}\right)^\mu = 1 + \sum_{n=1}^\infty \frac{L_n(b_0, b_1, \cdots, b_{n-1})}{z^n},$$

*where*

$$L_n(b_0, b_1, \cdots, b_{n-1}) = \sum_{i=0}^n K_{n-i}^\lambda(F_1, \cdots, F_{n-i}) K_i^\mu(b_0, \cdots, b_{i-1}) \qquad \left(K_0^\lambda = K_0^\mu = 1\right)$$

*and $F_n = F_n(b_0, b_1, \cdots, b_{n-1})$ is given by (5).*

**Proof.** By using Lemmas 1 and 2, we have

$$\left(\frac{zf'(z)}{f(z)}\right)^\lambda \left(\frac{f(z)}{z}\right)^\mu = \left(1 + \sum_{m=1}^\infty \frac{F_m(b_0, b_1, \cdots, b_{m-1})}{z^m}\right)^\lambda$$

$$\cdot \left(1 + \sum_{m=1}^\infty \frac{K_m^\mu(b_0, b_1, \cdots, b_{m-1})}{z^m}\right).$$

In addition, by applying Lemma 1 once again, we obtain

$$\left(\frac{zf'(z)}{f(z)}\right)^\lambda \left(\frac{f(z)}{z}\right)^\mu = \left(1 + \sum_{m=1}^\infty \frac{K_m^\lambda(F_1, \cdots, F_m)}{z^m}\right)$$

$$\cdot \left(1 + \sum_{m=1}^\infty \frac{K_m^\mu(b_0, \cdots, b_{m-1})}{z^m}\right)$$

$$= 1 + \sum_{n=1}^\infty \sum_{i=0}^n K_{n-i}^\lambda(F_1, \cdots, F_{n-i}) K_i^\mu(b_0, \cdots, b_{i-1}) \frac{1}{z^n}$$

$$\left(K_0^\lambda = K_0^\mu = 1\right).$$

Our demonstration of Lemma 3 is thus completed. □

The first three terms of $L_n$ are given by

$$L_1(b_0) = (\mu - \lambda)b_0,$$

$$L_2(b_0, b_1) = \frac{\lambda(1 + \lambda - 2\mu) + \mu(\mu - 1)}{2}b_0^2 + (\mu - 2\lambda)b_1$$

*and*

$$L_3(b_0, b_1, b_2) = \left(\frac{\lambda(2 - \mu)(\mu - \lambda)}{2} + \frac{\mu(\mu - 1)(\mu - 2) - \lambda(\lambda - 1)(\lambda - 2)}{6}\right)b_0^3$$

$$+ \left[\lambda(2\lambda + 1) + \mu(\mu - 3\lambda - 1)\right]b_0 b_1 + (\mu - 3\lambda)b_2.$$

**Remark 4.** *In the special case when $b_0 = b_1 = \cdots = b_{n-1} = 0$, we easily find that*

$$L_i(b_0, \cdots, b_{i-1}) = 0 \qquad (1 \leqq i \leqq n)$$

*and*

$$L_{n+1}(b_0, b_1, \cdots, b_n) = (\mu - (n+1)\lambda)b_n.$$

**Lemma 4** (see [37]). *If the function $p \in \mathcal{P}$, then $|c_k| \leqq 2$ for each $k$, where $\mathcal{P}$ is the family of all functions $p$, which are analytic in the domain $\Delta$ given by*

$$\Delta = \{z : z \in \mathbb{C} \quad and \quad 1 < |z| < \infty\}$$

*for which*

$$\Re(p(z)) > 0 \qquad (z \in \Delta),$$

*where*

$$p(z) = 1 + \frac{c_1}{z} + \frac{c_2}{z^2} + \frac{c_3}{z^3} + \cdots.$$

## 3. The Comprehensive Class $\Sigma_B(\lambda, \mu, \beta)$

In this section, we introduce and investigate the comprehensive class $\Sigma_B(\lambda, \mu, \beta)$ of meromorphic bi-univalent functions defined on the domain $\Delta$.

**Definition 2.** *A function $f \in \Sigma_\mathcal{M}$, given by* (2), *is said to be in the class*

$$\Sigma_B(\lambda, \mu, \beta) \quad (\lambda \geqq 1; \ \mu \geqq 0; \ 0 \leqq \beta < 1)$$

*of meromorphic bi-univalent functions of order $\beta$ and type $\mu$, if the following conditions are satisfied:*

$$\Re\left(\left(\frac{zf'(z)}{f(z)}\right)^\lambda \left(\frac{f(z)}{z}\right)^\mu\right) > \beta$$

*and*

$$\Re\left(\left(\frac{wg'(w)}{g(w)}\right)^\lambda \left(\frac{g(w)}{w}\right)^\mu\right) > \beta,$$

*where the function $g$ given by* (4), *is the inverse of $f$ and $z, w \in \Delta$.*

**Remark 5.** *There are several choices of the parameters $\lambda$ and $\mu$ which would provide interesting subclasses of meromorphic bi-univalent functions. For example, we have the following special cases:*

- *By putting $\lambda = 1$ and $0 \leqq \mu < 1$, the class $\Sigma_B(\lambda, \mu, \beta)$ reduces to the subclass $B(\beta, \mu)$ of meromorphic bi-Bazilevič functions of order $\beta$ and type $\mu$, which was considered by Jahangiri et al. [33].*
- *By putting $\lambda = 1$ and $\mu = 0$, the class $\Sigma_B(\lambda, \mu, \beta)$ reduces to the subclass $\Sigma_B^*(\beta)$ of meromorphic bi-starlike functions of order $\beta$, which was considered by Hamidi et al. [32].*
- *By putting $\mu = \lambda - 1$, the class $\Sigma_B(\lambda, \mu, \beta)$ reduces to the class $\Sigma_{B^*}(\lambda, \beta)$ in Definition 1.*

**Theorem 2.** *Let $f \in \Sigma_B(\lambda, \mu, \beta)$. If $b_0 = b_1 = \cdots = b_{n-1} = 0$, then*

$$|b_n| \leqq \frac{2(1-\beta)}{|(n+1)\lambda - \mu|} \qquad (n \geqq 1).$$

**Proof.** By using Lemma 3 for the meromorphic bi-univalent function $f$ given by

$$f(z) = z + b_0 + \sum_{n=1}^\infty \frac{b_n}{z^n},$$

we have

$$\left(\frac{zf'(z)}{f(z)}\right)^\lambda \left(\frac{f(z)}{z}\right)^\mu = 1 + \sum_{n=0}^\infty \frac{L_{n+1}(b_0, b_1, \cdots, b_n)}{z^{n+1}}. \tag{6}$$

Similarly, for its inverse map $g$ given by

$$g(w) = f^{-1}(w) = w + B_0 + \sum_{n=1}^{\infty} \frac{B_n}{w^n},$$

we find that

$$\left(\frac{wg'(w)}{g(w)}\right)^{\lambda}\left(\frac{g(w)}{w}\right)^{\mu} = 1 + \sum_{n=0}^{\infty} \frac{L_{n+1}(B_0, B_1, \cdots, B_n)}{w^{n+1}}. \tag{7}$$

Furthermore, since $f \in \Sigma_B(\lambda, \mu, \beta)$, by using Definition 2, there exist two positive real-part functions

$$c(z) = 1 + \sum_{n=1}^{\infty} c_n z^{-n}$$

and

$$d(w) = 1 + \sum_{n=1}^{\infty} d_n w^{-n}$$

for which

$$\Re\big(c(z)\big) > 0 \quad \text{and} \quad \Re\big(d(w)\big) > 0 \quad (z, w \in \Delta),$$

such that

$$\left(\frac{zf'(z)}{f(z)}\right)^{\lambda}\left(\frac{f(z)}{z}\right)^{\mu} = 1 + (1 - \beta)\sum_{n=0}^{\infty} K_{n+1}^1(c_1, c_2, \cdots, c_{n+1})\frac{1}{z^{n+1}} \tag{8}$$

and

$$\left(\frac{wg'(w)}{g(w)}\right)^{\lambda}\left(\frac{g(w)}{w}\right)^{\mu} = 1 + (1 - \beta)\sum_{n=0}^{\infty} K_{n+1}^1(d_1, d_2, \cdots, d_{n+1})\frac{1}{w^{n+1}}. \tag{9}$$

Upon equating the corresponding coefficients in (6) and (8), we get

$$L_{n+1}(b_0, b_1, \cdots, b_n) = (1 - \beta)K_{n+1}^1(c_1, c_2, \cdots, c_{n+1}). \tag{10}$$

Similarly, from (7) and (9), we obtain

$$L_{n+1}(B_0, B_1, \cdots, B_n) = (1 - \beta)K_{n+1}^1(d_1, d_2, \cdots, d_{n+1}). \tag{11}$$

Now, since $b_i = 0 \;\; (0 \leqq i \leqq n - 1)$, we have

$$B_i = 0 \quad (0 \leqq i \leqq n - 1) \quad \text{and} \quad B_n = -b_n.$$

Hence, by using Remark 4, Equations (10) and (11) can be rewritten as follows:

$$(\mu - (n + 1)\lambda)b_n = (1 - \beta)c_{n+1} \tag{12}$$

and

$$-(\mu - (n + 1)\lambda)b_n = (1 - \beta)d_{n+1}, \tag{13}$$

respectively. Thus, from (12) and (13), we find that

$$2(\mu - (n + 1)\lambda)b_n = (1 - \beta)(c_{n+1} - d_{n+1}).$$

Finally, by applying Lemma 4, we get

$$|b_n| = \frac{(1 - \beta)|c_{n+1} - d_{n+1}|}{2|(n + 1)\lambda - \mu|} \leqq \frac{2(1 - \beta)}{|(n + 1)\lambda - \mu|},$$

which completes the proof of Theorem 2　□

**Theorem 3.** *Let the function $f \in \mathcal{M}$, given by (2), be in the class*

$$\Sigma_B(\lambda, \mu, \beta) \quad (\lambda \geqq 1; \ \mu \geqq 0; \ 0 \leqq \beta < 1).$$

*Then,*

$$|b_0| \leqq \min\left\{ \frac{2(1-\beta)}{|\mu-\lambda|}, 2\sqrt{\frac{1-\beta}{|\lambda(1+\lambda-2\mu)+\mu(\mu-1)|}} \right\},$$

$$|b_1| \leqq \frac{2(1-\beta)}{|\mu-2\lambda|}$$

*and*

$$|b_2| \leqq \frac{2\{|\lambda(2\lambda+4)+\mu(\mu-3\lambda-2)|+|\lambda(2\lambda+1)+\mu(\mu-3\lambda-1)|\}(1-\beta)}{|(\mu-3\lambda)[\lambda(4\lambda+5)+\mu(2\mu-6\lambda-3)]|}$$
$$+ \frac{8|T(\mu,\lambda)|(1-\beta)^3}{|(\mu-3\lambda)(\mu-\lambda)^3|},$$

*where*

$$T(\mu,\lambda) = \frac{\lambda(2-\mu)(\mu-\lambda)}{2} + \frac{\mu(\mu-1)(\mu-2)-\lambda(\lambda-1)(\lambda-2)}{6}.$$

**Proof.** By putting $n = 0, 1, 2$ in (10), we get

$$(\mu-\lambda)b_0 = (1-\beta)c_1, \tag{14}$$

$$\frac{\lambda(1+\lambda-2\mu)+\mu(\mu-1)}{2}b_0^2 + (\mu-2\lambda)b_1 = (1-\beta)c_2 \tag{15}$$

and

$$T(\mu,\lambda)b_0^3 + [\lambda(2\lambda+1)+\mu(\mu-3\lambda-1)]b_0b_1 + (\mu-3\lambda)b_2 = (1-\beta)c_3. \tag{16}$$

Similarly, by putting $n = 0, 1, 2$ in (11), we have

$$-(\mu-\lambda)b_0 = (1-\beta)d_1, \tag{17}$$

$$\frac{\lambda(1+\lambda-2\mu)+\mu(\mu-1)}{2}b_0^2 - (\mu-2\lambda)b_1 = (1-\beta)d_2 \tag{18}$$

and

$$-T(\mu,\lambda)b_0^3 + (\lambda(2\lambda+4)+\mu(\mu-3\lambda-2))b_0b_1 - (\mu-3\lambda)b_2 = (1-\beta)d_3. \tag{19}$$

Clearly, from (14) and (17), we get

$$c_1 = -d_1 \tag{20}$$

and

$$b_0 = \frac{(1-\beta)c_1}{\mu-\lambda}. \tag{21}$$

Adding (15) and (18), we obtain

$$b_0^2 = \frac{(1-\beta)(c_2+d_2)}{\lambda(1+\lambda-2\mu)+\mu(\mu-1)}. \tag{22}$$

In view of the Equations (21) and (22), by applying Lemma 4, we get

$$|b_0| \leqq \frac{2(1-\beta)}{|\mu - \lambda|} \qquad \text{and} \qquad |b_0|^2 \leqq \frac{4(1-\beta)}{|\lambda(1 + \lambda - 2\mu) + \mu(\mu - 1)|},$$

respectively. Thus, we get the desired estimate on the coefficient $|b_0|$.

Next, in order to find the bound on the coefficient $|b_1|$, we subtract (18) from (15). We thus obtain

$$b_1 = \frac{(1-\beta)(c_2 - d_2)}{2(\mu - 2\lambda)}. \tag{23}$$

Applying Lemma 4 once again, we get

$$|b_1| \leqq \frac{2(1-\beta)}{|\mu - 2\lambda|}.$$

Finally, in order to determine the bound on $|b_2|$, we consider the sum of the Equations (16) and (19) with $c_1 = -d_1$. This yields

$$b_0 b_1 = \frac{(1-\beta)(c_3 + d_3)}{\lambda(4\lambda + 5) + \mu(2\mu - 6\lambda - 3)}. \tag{24}$$

Subtracting (19) from (16) with $c_1 = -d_1$, we obtain

$$2(\mu - 3\lambda)b_2 + (\mu - 3\lambda)b_0 b_1 + 2T(\mu, \lambda)b_0^3 = (1-\beta)(c_3 - d_3). \tag{25}$$

In addition, by using (21) and (24) in (25), we get

$$b_2 = \frac{(1-\beta)(c_3 - d_3)}{2(\mu - 3\lambda)} - \frac{(1-\beta)(c_3 + d_3)}{2[\lambda(4\lambda + 5) + \mu(2\mu - 6\lambda - 3)]} - \frac{T(\mu, \lambda)(1-\beta)^3 c_1^3}{(\mu - 3\lambda)(\mu - \lambda)^3}.$$

Hence,

$$b_2 = \frac{\{[\lambda(2\lambda + 4) + \mu(\mu - 3\lambda - 2)]c_3 - [\lambda(2\lambda + 1) + \mu(\mu - 3\lambda - 1)]d_3\}(1-\beta)}{(\mu - 3\lambda)[\lambda(4\lambda + 5) + \mu(2\mu - 6\lambda - 3)]}$$
$$- \frac{T(\mu, \lambda)(1-\beta)^3 c_1^3}{(\mu - 3\lambda)(\mu - \lambda)^3}.$$

Thus, by applying Lemma 4 once again, we get

$$|b_2| \leqq \frac{2\{|\lambda(2\lambda + 4) + \mu(\mu - 3\lambda - 2)| + |\lambda(2\lambda + 1) + \mu(\mu - 3\lambda - 1)|\}(1-\beta)}{|(\mu - 3\lambda)[\lambda(4\lambda + 5) + \mu(2\mu - 6\lambda - 3)]|}$$
$$+ \frac{8|T(\mu, \lambda)|(1-\beta)^3}{|(\mu - 3\lambda)(\mu - \lambda)^3|}.$$

This completes the proof of Theorem 3. $\square$

## 4. A Set of Corollaries and Consequences

By setting $\lambda = 1$ and $0 \leqq \mu < 1$ in Theorem 2, we have the following result.

**Corollary 1.** *Let the function $f \in \mathcal{M}$, given by (2), be in the subclass $B(\beta, \mu)$ of meromorphic bi-Bazilevič functions of order $\beta$ and type $\mu$. If*

$$b_0 = b_1 = \cdots = b_{n-1} = 0,$$

*then*

$$|b_n| \leqq \frac{2(1-\beta)}{n + 1 - \mu} \qquad (n \geqq 1).$$

**Remark 6.** *The estimate of $|b_n|$, given in Corollary 1, is the same as the corresponding estimate given by Hamidi et al. [38] Corollary 3.3.*

By setting $\mu = 0$ in Corollary 1, we have the following result.

**Corollary 2.** *Let the function $f \in \mathcal{M}$, given by (2), be in the subclass $\Sigma_B^*(\beta)$ of meromorphic bi-starlike functions of order $\beta$. If*

$$b_0 = b_1 = \cdots = b_{n-1} = 0,$$

*then*

$$|b_n| \leqq \frac{2(1-\beta)}{n+1} \qquad (n \geqq 1).$$

**Remark 7.** *The estimate of $|b_n|$, given in Corollary 2, is the same as the corresponding estimate given by Hamidi et al. [38] Corollary 3.4.*

By setting $\mu = \lambda - 1$ in Theorem 2, we have the following result.

**Corollary 3.** *Let the function $f \in \mathcal{M}$, given by (2), be in the subclass $\Sigma_{B*}(\lambda, \beta)$. If*

$$b_0 = b_1 = \cdots = b_{n-1} = 0,$$

*then*

$$|b_n| \leqq \frac{2(1-\beta)}{n\lambda + 1} \qquad (n \geqq 1).$$

**Remark 8.** *Corollary 3 is a generalization of a result presented in Theorem 1, which was proved by Srivastava et al. [31].*

By setting $\lambda = 1$ and $0 \leqq \mu < 1$ in Theorem 3, we have the following result.

**Corollary 4.** *Let the function $f \in \mathcal{M}$, given by (2), be in the subclass $B(\beta, \mu)$ of meromorphic bi-Bazilevič functions of order $\beta$ and type $\mu$. Then,*

$$|b_0| \leqq \begin{cases} \sqrt{\frac{4(1-\beta)}{(1-\mu)(2-\mu)}} & \left(0 \leqq \beta \leqq \frac{1}{2-\mu}\right) \\[2ex] \frac{2(1-\beta)}{1-\mu} & \left(\frac{1}{2-\mu} \leqq \beta < 1\right), \end{cases}$$

$$|b_1| \leqq \frac{2(1-\beta)}{2-\mu}$$

*and*

$$|b_2| \leqq \frac{2(1-\beta)}{3-\mu} + \frac{4(2-\mu)(1-\beta)^3}{3(1-\mu)^2}.$$

**Remark 9.** *Corollary 4 also contains the estimate of the Taylor–Maclaurin coefficient $|b_2|$ of functions in the subclass $B(\beta, \mu)$ (see [33]).*

By setting $\mu = 0$ in Corollary 4, we have the following result.

**Corollary 5.** *Let the function $f \in \mathcal{M}$, given by (2), be in the subclass $\Sigma_B^*(\beta)$ of meromorphic bi-starlike functions of order $\beta$. Then,*

$$
|b_0| \leqq
\begin{cases}
\sqrt{2(1-\beta)} & \left(0 \leqq \beta \leqq \tfrac{1}{2}\right) \\[2mm]
2(1-\beta) & \left(\tfrac{1}{2} \leqq \beta < 1\right),
\end{cases}
$$

$$
|b_1| \leqq 1 - \beta
$$

*and*

$$
|b_2| \leqq \frac{2(1-\beta)}{3} + \frac{8(1-\beta)^3}{3}.
$$

**Remark 10.** *Corollary 5 not only improves the estimate of the Taylor–Maclaurin coefficient $|b_0|$, which was given by Hamidi et al. [32] Theorem 2, but it also provides an improvement of the known estimate of the Taylor–Maclaurin coefficient $|b_2|$ of functions in the subclass $\Sigma_B^*(\beta)$. Furthermore, the estimate of $|b_0|$, presented in Corollary 5, is the same as the corresponding estimate given by Hamidi et al. [38] Corollary 3.5.*

By setting $\mu = \lambda - 1$ in Theorem 3, we have the following result.

**Corollary 6.** *Let the function $f \in \mathcal{M}$, given by (2), be in the subclass $\Sigma_{B^*}(\lambda, \beta)$. Then,*

$$
|b_0| \leqq
\begin{cases}
\sqrt{2(1-\beta)} & \left(0 \leqq \beta \leqq \tfrac{1}{2}\right) \\[2mm]
2(1-\beta) & \left(\tfrac{1}{2} \leqq \beta < 1\right),
\end{cases}
$$

$$
|b_1| \leqq \frac{2(1-\beta)}{\lambda + 1}
$$

*and*

$$
|b_2| \leqq \frac{2(1-\beta)}{2\lambda + 1} + \frac{8(1-\beta)^3}{2\lambda + 1}.
$$

**Remark 11.** *Corollary 6 improves the estimates of the Taylor–Maclaurin coefficients $|b_0|$ and $|b_1|$ in Theorem 1 of Srivastava et al. [31]. In fact, it also provides an improvement of the known estimate of the Taylor–Maclaurin coefficient $|b_2|$ of functions in the subclass $\Sigma_{B^*}(\lambda, \beta)$.*

**Remark 12.** *In his recently-published survey-cum-expository review article, Srivastava [39] demonstrated how the theories of the basic (or q-) calculus and the fractional q-calculus have significantly encouraged and motivated further developments in Geometric Function Theory of Complex Analysis (see, for example, [8,40–42]). This direction of research is applicable also to the results which we have presented in this article. However, as pointed out by Srivastava [39] (p. 340), any further attempts to easily (and possibly trivially) translate the suggested q-results into the corresponding $(p, q)$-results (with $0 < |q| < p \leqq 1$) would obviously be inconsequential because the additional parameter p is redundant.*

**Author Contributions:** All three authors contributed equally to this investigation. All authors have read and agreed to the published version of the manuscript.

**Funding:** This research received no external funding.

**Institutional Review Board Statement:** Not applicable.

**Informed Consent Statement:** Not applicable.

**Data Availability Statement:** Not applicable.

**Conflicts of Interest:** The authors declare no conflict of interest.

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
