# Peer review of "Coefficients of a Comprehensive Subclass of Meromorphic Bi-Univalent Functions Associated with the Faber Polynomial Expansion"

_axioms, doi:10.3390/axioms10010027_

Round 1
Reviewer 1 Report
The object of the submitted paper is to investigate some classes of meromorphic bi-univalent functions associated with Faber polynomial expansion. The authors obtain estimations for Taylor-Maclaurin coefficients. This manuscript is of technical character but the obtained result are interesting. They generalize or/and improve several earlier results.
Therefore, I recommend the paper for publication.
Author Response
We have revised the paper in accordance with your suggestions. Thank you for your careful reading of the paper and for your constructive review.
Reviewer 2 Report
The paper is in general well written and the results seem to me interesting. I recommend publication after minor changes:
1 Pag 2. l 20 Change "we suppose that "\Sigma" be the..., by Let $\Sigma$ be...
2 Give a reference, or at least a sketch of the argument, for the Laurent development of the inverses of $\Sigma$ class in page 2
3 Statements of Lemma 1, Lemma 2 and Lemma 3. Put "Let $f$ be the function..." (instead of "Let the function $f$...")
4 Theorem 3 and Corollary 1, Corollary 2, Corollary 4 and Corollary 5. "Let $f\in \Sigma_M$, given by (2), be a function in the class
5 Proof of Theorem 3. at the end. Avoid the last sentence, "Which evidently gives the proof". Put simply, "and hence we conclude", "we are done", "this completes the proof", there are a lot of standard expressions.
Author Response
We have revised the paper in accordance with your suggestions. You can find it in attachment. Thank you for your careful reading of the paper and for your constructive review.
